# Combined Analysis of Volatile Compounds and Extraction of Floral Fragrance Genes in Two *Dendrobium* Species

Yanni Yang [1], Ke Xia [1], Qiaofen Wu [1], Xi Lu [1], Shunjiao Lu [2], Zhiguo Zhao [1] and Shuo Qiu [1,*]

[1] Guangxi Key Laboratory of Plant Functional Phytochemicals and Sustainable Utilization, Guangxi Institute of Botany, Guangxi Zhuang Autonomous Region and Chinese Academy of Sciences, Guilin 541006, China; yangyanni219@126.com (Y.Y.); xiake4502@163.com (K.X.); wuqfmj@163.com (Q.W.); luluxi_97@163.com (X.L.); 13788588632@139.com (Z.Z.)

[2] Tropical Crops Genetic Resources Institute, Chinese Academy of Tropical Agricultural Sciences, Haikou 571101, China; lushunjiao@catas.cn

[*] Correspondence: qiushuo001@163.com; Tel.: +86-773-3550103; Fax: +86-773-3550067

**Abstract:** Many species of the *Dendrobium genus* are traditional Chinese herbal medicine and ornamental plants. Flower fragrance is one of the most important horticultural ornamental characters and plays a crucial role in the ecology, economy, and aesthetics of plants. However, the volatile constituents and key regulatory genes related to floral biosynthesis are poorly understood. In this experiment, the flowers from two species of *Dendrobium* with high-scent smells, *Dendrobium moniliforme* (L.) Sw. (*D. moniliforme*), and light-scent smells, *Dendrobium nobile* "H1" (*D.* "H1"), were selected. The aim of this study was to explore the key gene expression profiles of floral biosynthesis by combining volatile constituent determination and transcriptome analysis in two different *Dendrobium* species. Physiological determination results showed that 60 volatile compounds were identified in *D. moniliforme* and 52 volatile compounds were identified in *D.* 'H1' flowers in four flowering stages, and the full bloom stage was the most complicated stage because there were 41 and 33 volatile compounds, respectively. These compounds belong to terpenes, aromatics, fatty acids, nitrogenous compounds, ketones, alcohols, and alkanes, respectively. The components identified in the gynandrium and petals revealed that the petals were probably the most important sites affecting the release of volatiles. The relative content of terpene compounds was the highest, with 77.25% (*D. moniliforme*) and 50.38% (*D.* "H1"), respectively. Transcriptome analysis showed that differentially expressed genes (DEGs) were highly enriched in terpenoid backbone biosynthesis and that four linalool synthetase (LIS) genes were up-regulated in high-scent smell species. This study is helpful to explore the key genes of flower fragrance and provides a theoretical basis for further understanding of the regulatory molecular functions of floral synthesis and release, as well as for the cultivation of new aromatic species.

**Keywords:** *Dendrobium*; volatile compounds; transcriptome; floral metabolites; terpenoid

## 1. Introduction

*Dendrobium* genus belongs to the orchidaceae herbs, which make up one of the largest families of flowering plants with abundant ecological habitats and extensive economic impacts for their ornamental and medicinal values in the world [1,2]. There are more than 74 species and 2 variant *Dendrobium* species in China [3], of which only a few can release their flower fragrance [4]. Floral fragrances not only attract insects and other pollinators to help plants reproduce [5], driving away natural enemies for self-protection and resisting abiotic stress [6], but are also one of the most important horticultural ornamental characters of *Dendrobium* [7].

The main volatile compounds of *Dendrobium* included terpenoids, aldehydes, esters, and alcohols [8]. Among them, terpenoids are one of the most abundant and diverse aromatic compounds in flowers [9,10]. In recent years, more than 80,000 terpenoids have been identified, and the high volatility of terpenoid compounds promotes the fragrance

of *Dendrobium* [11–15]. Terpenoids have diverse biological functions in nature and play an essential role in plant-animal [16,17], plant-pathogen [9,18], and plant–plant interactions and the regulation of antagonism between organisms [19]. There are two main metabolic pathways for forming terpenoids: the mevalonate pathway (MVA pathway) and the 2C-methyl-D-erythritol-4-phosphate pathway (MEP pathway). Both pathways (MVA and MEP) are responsible for forming isoprene, which is the building block of various types of terpenes [20]. All-trans-farnesyl diphosphate (FPP) is transformed into a sesquiterpene skeleton by the MVA pathway, while geranyl diphosphate (GPP) and all-trans-geranylgeranyl diphosphate (GGPP) are produced as monoterpene and diterpene skeletons by the MEP, respectively [21]. TPS is the final enzyme converting the precursors FPP, GPP, and GGPP to different kinds of sesquiterpene, monoterpenes, and diterpene. In addition, TPS harbors conserved motif structures such as DDxxD (an aspartate-rich motif that interacts with divalent metal ions involved in positioning the substrate for catalysis) in plants [22], but their catalytic activity and products vary greatly. Structural differences may lead to new catalytic activities, resulting in the formation of many different species of volatile terpenoids from the same substrate [23]. Monoterpenes and sesquiterpenes are the majority of volatile compounds [18]. TPSs are highly differentiated gene families that have been found in *Arabidopsis thaliana*, *Solanum lycopersicum*, and *Vitis vinifera* [24]. In recent years, genome-wide analyses of TPSs have identified them in plants such as *Gossypium hirsutum* [25], *Brachypodium distachyon* [26], and *Liriodendron chinense* [27]. Bioinformatics techniques were used to compare plant TPS family members, which can be divided into 8 subfamilies: TPSa (sesquiterpenes), TPSb (cyclic monoterpenes and hemiterpenes), TPSc (copalyl diphospate synthases), TPSd (gymnosperm-specific), TPSe (ent-kaurene synthases), TPSf (other diterpene synthases), TPSg (acyclic monoterpenes), and TPSh (lycopod-specific) [28].

The volatile compounds of orchids contain various terpenes. Monoterpenes, such as linalool, pinene, ylangene, and limonene, are major floral scent compounds in Orchid flowers [20,29–31]. Among the monoterpene synthetase genes, linalool synthetase (LIS), geraniol synthetase (GES), and β-Basil synthetase (OS) have been studied more, which belong to the TPSg family [22,32]. To date, only a few TPS genes have been identified in orchids. A total of 34 TPS genes were identified In *Dendrobium officinale*, and these genes were mainly expressed in the flowers, followed by the roots and stems [24]. PbTPS5 and PbTPS10 genes are involved in monoterpene biosynthesis in *Phalaenopsis bellina* [33]. In the flowers of the *Freesia hybrid*, a total of 8 TPS genes were identified. Among them, FhTPS1 catalyzed the formation of linalool, whereas FhTPS4, FhTPS6, and FhTPS7 were poly-product enzymes that could recognize both substrates, GPP and FPP [34].

There are many species of *Dendrobium*, but few have flowers that are both fragrant and beautiful. In the experiment, we selected two species of *Dendrobium*: *Dendrobium moniliforme* (L.) Sw. (*D. moniliforme*) and *Dendrobium nobile* "H1" (*D.* "H1"). *D. moniliforme* is a wild germplasm widely distributed in the south of China, the Korea Peninsula, northeastern India, and Japan [35]. It grows on tree trunks, valley rock walls, or even cliffs. It has the characteristics of thin stems, small flowers, and a high-scent fragrance of flowers. Unlike *D. moniliforme*, *D.* "H1" is a cultivated line with large and bright flowers but only a faint fragrance. By combining volatile constituent determination and transcriptome sequencing analysis of these two different *Dendrobium* species, the characteristics of volatile constituents in different florescences and organs were analyzed, and the key genes involved in terpenoid synthesis were explored. Furthermore, real-time quantitative PCR (qRT-PCR) was used for the detection of the correlation between transcriptome sequencing and gene expression levels.

## 2. Materials and Methods

### 2.1. Plant Materials

In this experiment, the fresh flowers of light-scent *D.* "H1" and high-scent *D. moniliforme* were used as materials, both of which were obtained from the laboratory of Professor

Shuo Qiu of the Guangxi Institute of Botany, complying with Chinese legislation (Figure 1). *D.* "H1" was bought from Zhejiang Senhe Seed Co., Ltd. (Hangzhou City, Zhejiang Province, China), and *D. moniliforme* came from Baise City, Guangxi Province, China. Twelve flower samples were collected from each variety in different florescence stages and different organs, with three replicates for each of the two species, to determine the natural volatile constituents. The petals at the full bloom stage were chosen for transcriptome analysis. Using solid phase microextraction (SPME) and gas chromatography coupled with mass spectrometry (GC-MS) to determine the natural volatile constituents of these two *Dendrobium* species in different florescence stages (bud stage, first flowering stage, full bloom stage, and declining stage) and flower parts (gynandrium and petal) [36,37]. To investigate the temporal and spatial correlation between volatile compound-related genes, according to the results of physiological determination, we collected petals at full bloom stage, including two species of flowers, for RNA extraction. All samples were immediately frozen in liquid nitrogen and stored at −80 °C until needed.

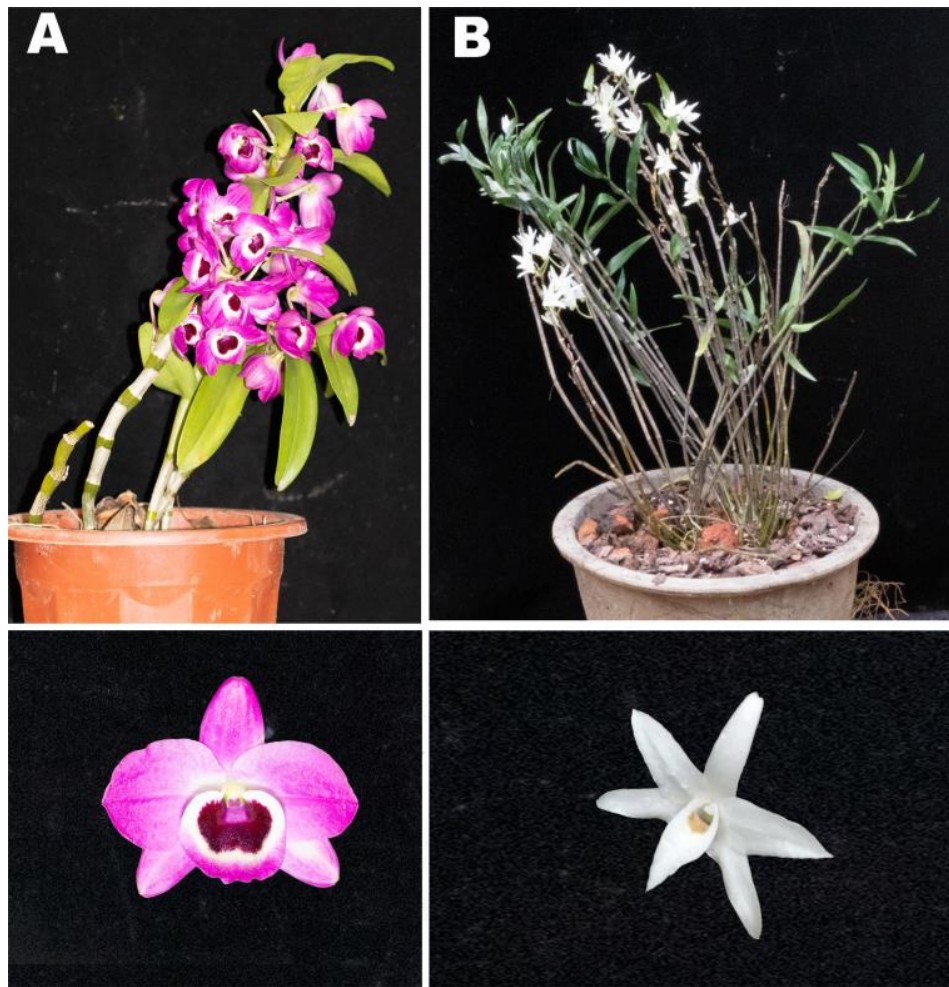

**Figure 1.** Two species of *Dendrobium* flowers. (**A**) *D.* "H1". (**B**) *D. moniliforme*.

*2.2. Apparatus*

Manual solid phase micro extraction injector and 50/30 μm PDMS/CAR/DVB extractor (SUPELCO, Inc., Bellefonte, PA, USA), 6890N-5975B Gas chromatography-mass spectrometer (GC-MS) apparatus (Agilent Technologies, Santa Clara, CA, USA), 40 mL brown headspace sampling bottle, and water bath (Shanghai Jingxue Scientific Instruments Co., Ltd., Shanghai, China).

### 2.3. GC-MS Analysis

For the analysis of natural volatile compounds, *D. moniliforme* and *D.* "H1" fresh flowers were enclosed, sampled in brown headspace sampling bottles, and repeated three times. The extractor was inserted into the GC-MS injector for 30 min at 250 °C. Fiber heads were inserted in brown headspace sampling bottles containing 12 flowers, and headspace extraction was performed at 40 °C for 30 min. After extraction, the fiber head was removed and inserted into the GC-MS injection port. After analysis for 5 min, the sample was injected for further analysis [38].

The flow rate of the HP5-MS quartz capillary column (30 m × 0.25 mm × 0.25 μm) was 0.8 mL·min$^{-1}$. The carrier gas was high-purity helium (99. 999%), and splitless mode was selected. Program temperature setting: the initial column temperature was at 40 °C for 3 min, then increased at a rate of 3 °C min$^{-1}$ to 73 °C for 3 min, and then the temperature was heated to 220 °C at 5 °C·min$^{-1}$ for 2 min. The temperature of the electron ionization (EI) ion source was maintained at 230 °C, and the electron energy was 70 eV. The temperature of the GC-MS transmission line was 250 °C, and the scanning range was 40–450 amu [39].

According to the total ionization chromatography of GC-MS and to determine the volatile compounds detected during the experiment, the National Institute of Standards and Technology, https://www.nist.gov/ (accessed on 18 March 2023) was searched and compared with the standard spectrum of the Eight Peaks Index and the EPA/NIH mass spectral data base, respectively [40]. Using the Xcalibur1.2 software, quantitative analysis was carried out according to the peak area normalization method, and the phase pair contents of each chemical component were obtained, respectively.

### 2.4. Transcriptome Analysis

The TRlzol Reagent (Life Technologies, Carlsbad, CA, USA) was used for the total RNA extraction of each sample. Using the Agilent 2100 Bioanalyzer (Agilent Technologies, Santa Clara, CA, USA), checked RNA integrity and concentration. The mRNA was isolated by the NEBNext Poly (A) mRNA Magnetic Isolation Module (NEB, E7490). The Illumina NEBNext Ultra RNA Library Prep Kit (NEB, E7530) and NEBNext Multiplex Oligos (NEB, E7500) were used in the cDNA library construction [41]. Finally, the constructed cDNA libraries were sequenced on a flow cell on the Illumina HiSeq™ sequencing platform.

Low quality and shorter reads were removed by the Perl script. The clean reads that were filtered from the raw reads were mapped to the *Dendrobium* genome (OGSv3.2) using Tophat2 software. The transcriptome data have been deposited into the NCBI Sequence Read Archive with the identifier PRJNA976822. Then gene expression levels were calculated using FPKM values (fragments per kilobase of exon per million fragments mapped) by the Cufflinks software. DESeq and Q-value were employed and used to evaluate differential expression genes (DEGs) between *D.* 'H1' and *D. moniliforme*. After that, gene abundance differences between those samples were calculated based on the ratio of the FPKM values [42].

To compute the significance of the differences, only genes with a *p*-value of log2 ratio ≥ 2 and a false discovery rate (FDR) significance score < 0.01 were used for subsequent analysis. With a cut-off E-value of $10^{-5}$, genes were compared against various protein databases by BLASTX, using the National Center for Biotechnology Information (NCBI) non-redundant protein (Nr) database and Swiss-Prot database [43]. Furthermore, the Nr BLAST results were imported into the Blast2 GO program to obtain the gene ontology (GO) term annotation genes [44]. Then, a Perl script was used to plot GO functional classification, and KEGG pathways were assigned to the assembled sequences for the unigenes [45]. The resulting comments were enriched and refined using TopGo (the R package) [46]. Gene sequences were also compared with the Cluster Homologous Group (COG) database to predict and classify function [47].

*2.5. RT-PCR Analysis*

According to the physiological determination, combined transcriptome sequencing analysis were performed to identify the differentially expressed genes related to floral biosynthesis and their expression levels were verified. Fresh *D.* "H1" and *D. moniliforme* flowers in the full bloom stage of petals were taken for qRT-PCR analysis. HUAYUEYANG RNA extraction kit (HUAYUEYANG Biotechnology, Beijing, China) was used to isolate total RNA from the samples. According to the manufacturer's protocol, 1 μg total RNA was used to synthesize cDNA using the Takara cDNA Synthesis Kit (biological engineering company, Dalian, China), and fluorescent quantitative primers were designed using primer 5.0 (Supplementary Table S5). Using the Roche Lightcycler 480 Real-time PCR System (Roche, Switzerland) and the SYBR Green PCR Reaction Master Mix Kit (Vazyme, Nanjing, China), the expression level of genes was detected by qRT-PCR. Procedure: 95 °C pre-denaturation for 3 min, then 95 °C reaction for 10 s, and finally 60 °C reaction for 30 s, 45 cycles. Each sample was repeated three times, and the relative expression level of each gene was calculated using $2^{-\Delta\Delta CT}$.

## 3. Results

*3.1. Analysis the Characteristics of Volatile Constituents in Different Florescences of Two Dendrobium Species*

According to the SPME and GC-MS analysis of volatile compounds from *D. moniliforme* and *D.* "H1", a total of seven main chemical groups, including terpenes, aromatics, fatty acids, nitrogenous compounds, ketone, alcohols, and alkanes, were collected. However, there were significant differences in the compositions and content of volatile compounds in the two *Dendrobium* species. As shown in Figure 2, terpenoids are the main components of the aromatic compounds in the flowers at the four flowering stages. The contents of terpenoids in *D. moniliforme* were 71.78% at the bud stage, 89.83% at the first flowering stage, 76.69% at the full bloom stage, and 64.33% at the declining stage. Otherwise, the terpenoids contents of *D.* "H1" in the four stages are 53.27%, 89.74%, 79.09%, and 35.78%, respectively. Among the 60 volatile compounds identified in *D. moniliforme*, 1R-α-pinene and Linalool were found in all four flowering stages. The relative contents of 1R-α-pinene and Linalool accounted for more than 40% of all compounds, which was the highest among all compounds in the four flowering stages (Supplementary Table S1). On the other hand, we found 52 volatile compounds in the flowers of *D.* "H1", and the relative contents of 1R-α-pinene were more than 21.11%, which was detected in four flowering stages. Except for the bud stage, the content of cis-β-ocimene was also relatively high (10.36–54%) (Supplementary Table S2). These results indicated the composition of volatile compounds becomes more complex as the flowers open, with the most complicated in the full bloom stage, which has 41 (*D. moniliforme*) and 33 (*D.* "H1") volatile compounds, respectively. In particular, the emissions of Linalool (one of the terpenoids) were higher in *D. moniliform* throughout all the flowering stages, which accounted for 10.21–27.84% of the total volatiles in *D. moniliform* but for only 0–1.37% in *D.* "H1". The difference in linalool content may be one of the main volatile components that distinguish floral fragrance release between these two *Dendrobium* species.

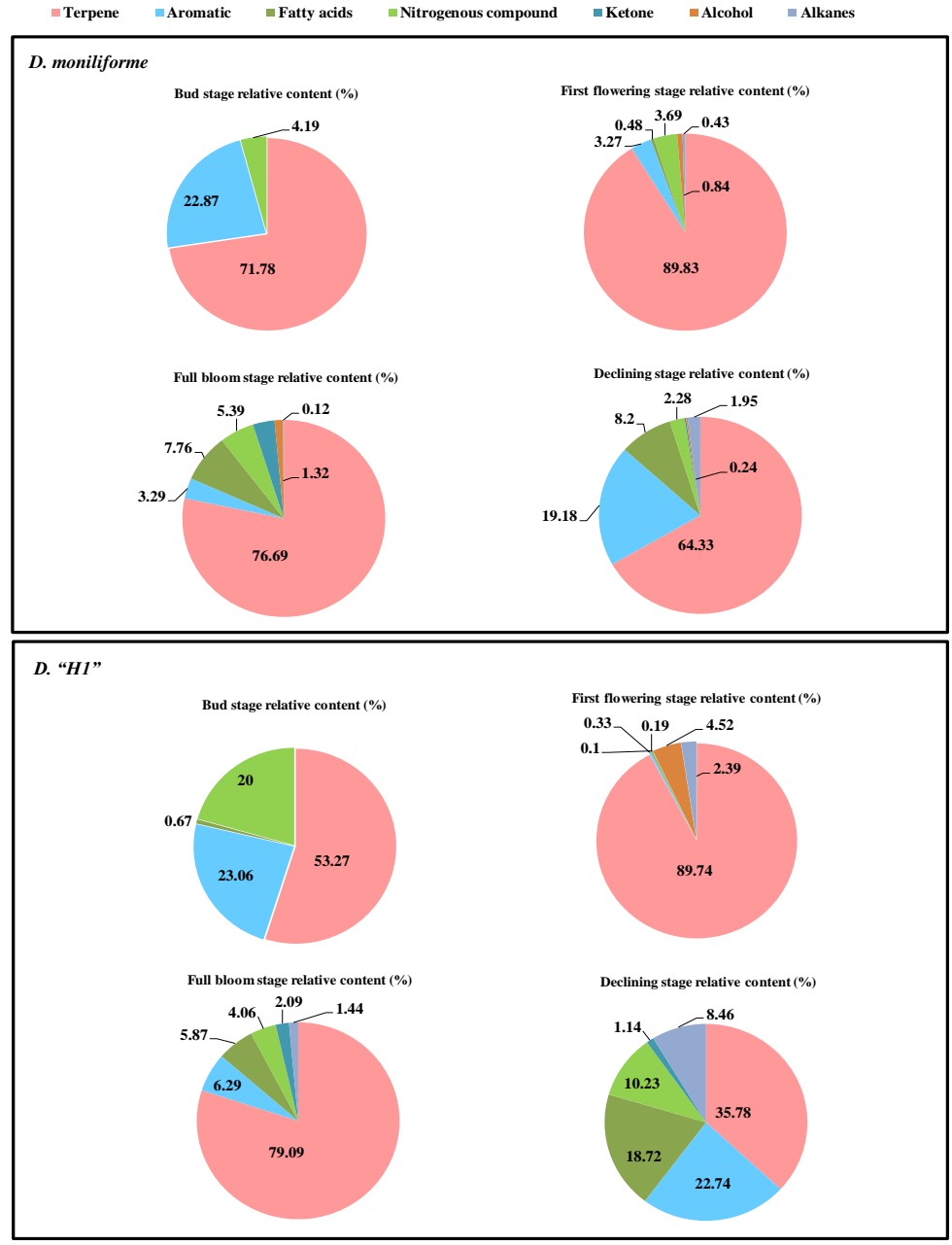

**Figure 2.** Volatile constituents in different florescences of two *Dendrobium* species.

*3.2. Analysis of the Characteristics of Volatile Constituents in Different Organs of Two Dendrobium Species*

By GC/MS analysis, the components and relative contents of the same compounds in different flower parts of the two *Dendrobium* species at full bloom stage were measured (Table 1). The highest relative content in the petals was still the terpenoids contents, and there were 15 kinds of terpenoids identified in *D. moniliforme* with a relative total content of 77.25%. *D.* 'H1' contained 13 kinds of terpenoids with a relative total content of 50.38%. Although there were only 7 kinds of terpenoids in the gynandrium of the two species, the relative total contents reached 81.75% (*D. moniliforme*) and 63.31% (*D.* 'H1'), respectively.

**Table 1.** Classification of scent compositions of different fluorescence in two flower parts of two *Dendoubium* species.

| Cultivar and Flower Part | Compound Name | Variety | Relative Content (%) |
|---|---|---|---|
| *D. moniliforme* Petals | Terpene | 15 | 77.25 |
| | Aromatic | 2 | 2.12 |
| | Fatty acids | 2 | 9.99 |
| | Nitrogenous compound | 6 | 4.94 |
| | Ketone | 0 | 0 |
| | Alcohol | 1 | 1.94 |
| | Alkanes | 0 | 0 |
| | Total | 26 | 96.24 |
| *D. moniliforme* Gynandrium | Terpene | 7 | 81.75 |
| | Aromatic | 3 | 5.8 |
| | Fatty acids | 0 | 0 |
| | Nitrogenous compound | 4 | 3.03 |
| | Ketone | 3 | 6.19 |
| | Alcohol | 0 | 0 |
| | Alkanes | 0 | 0 |
| | Total | 17 | 96.77 |
| *D.* H1 Petals | Terpene | 13 | 50.38 |
| | Aromatic | 7 | 18.16 |
| | Fatty acids | 4 | 11.24 |
| | Nitrogenous compound | 7 | 13.1 |
| | Ketone | 1 | 1.5 |
| | Alcohol | 1 | 0.58 |
| | Alkanes | 3 | 2.88 |
| | Total | 36 | 97.84 |
| *D.* 'H1' Gynandrium | Terpene | 7 | 63.31 |
| | Aromatic | 4 | 18.12 |
| | Fatty acids | 1 | 3.36 |
| | Nitrogenous compound | 4 | 7.58 |
| | Ketone | 1 | 1.29 |
| | Alcohol | 1 | 1.67 |
| | Alkanes | 1 | 2.21 |
| | Total | 19 | 97.54 |

The results in Tables S1 and S2 showed that at the full flowering stage, the volatile compounds of *D. moniliforme* and *D.* "H1" in petals were 26 and 36, respectively, and 17 and 19 volatile compounds in gynandrium, respectively. Thus, the volatile compounds of the petal are more complex than those of the gynandrium. These results suggested that the petals may be the most important part affecting the release of volatiles.

### 3.3. Identification and Analysis of the DEGs

Based on the determination of physiological composition, petals in full bloom were selected for RNA-seq analysis of differentially expressed genes (DEGs) in the two *Dendrobium* species. In three independent experiments, 24,199 genes were matched to the *Dendrobium* genome, of which 16,735 were expressed. Of these expressed genes, 841 genes were unique to *D.* 'H1', 1912 genes were unique to *D. moniliforme,* and 13,982 genes were found in both species (Figure 3A). DESeqR and Q-value were applied to determine if the genes in *D.* 'H1' and *D. moniliforme* species were significantly different based on 16,735 expressed genes with a 1% false discovery rate (FDR). According to fold change (FC) > 1 and $p < 0.01$, 5790 differentially expressed genes (DEGs) showed dynamic changes between *D.* "H1" and *D. moniliforme*. If compared with *D.* "H1", 3210 DEGs were up-regulated, and 2580 DEGs were down-regulated in *D. moniliforme* (Figure 3B).

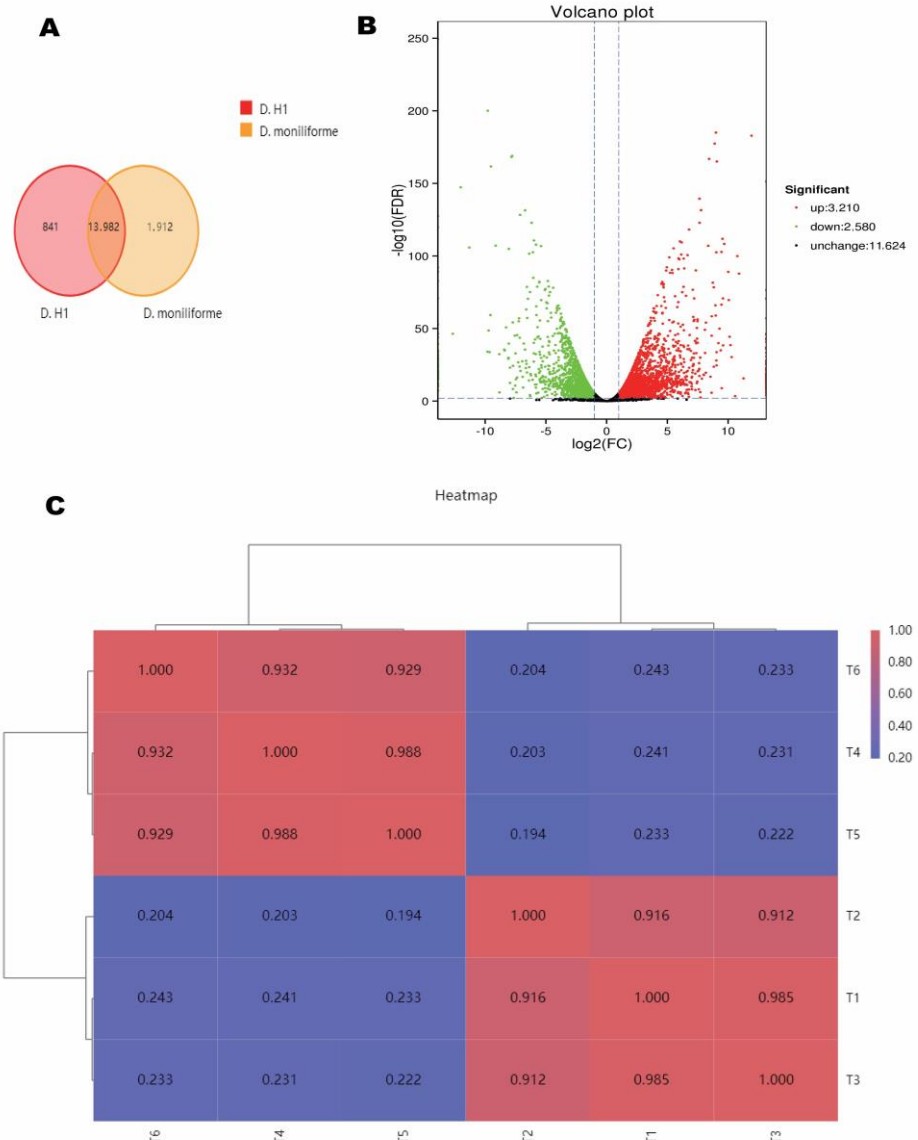

**Figure 3.** General information about the identified genes. (**A**) The distribution of expressed genes. The red circle were unique to *D.* "H1" and the yellow circle were unique to *D. moniliforme*, with genes found in both species shown at the intersection of the circles; (**B**) Venn diagram showing quantitative distribution of differentially expressed genes of the *D.* "H1" and *D. moniliforme*. The red dots represent upregulated differentially expressed genes, the green dots represent downregulated differentially expressed genes; (**C**) Hierarchical clustering analysis of the *D.* "H1" and *D. moniliforme*. T1–T3 represented the three replicates of *D.* "H1", and T4–T6 represented the three replicates of *D. moniliforme*.

Using the pHEATMAP package in R, we performed hierarchical clustering to show a comprehensive overview of the two species (Figure 3C). These results indicated that the three replicates of each sample from each species belong to the same respective clades, demonstrating the reproducibility and reliability of our RNA-Seq data.

### 3.4. KEGG Functional Classifications and Enrichment Analysis of the DEGs

The Kyoto Encyclopedia of Genes and Genomes (KEGG) functional classification analysis showed that the DEGs were mainly divided into five pathways: cellular processes, environmental information processing, genetic information processing, metabolism, and organismal systems. Among them, 932 unigenes were involved in the metabolism pathway, which was the most annotated gene in all pathways (Figure 4A). To further reveal the

metabolic pathways and functions of the identified DEGs, KEGG pathway enrichment analysis was performed (Figure 4B,C). The up-regulated genes were significantly enriched in plant hormone signal transduction, phenylpropanoid biosynthesis, and terpenoids backbone biosynthesis. The down-regulated proteins were significantly enriched in Glycolysis/Gluconeogenesis, fatty acid metabolism, and fatty acid biosynthesis. Based on the number of genes, these DEGs from *D.* "H1" and *D. moniliforme* were highly associated with phenylpropanoid biosynthesis (49 unigenes involved), terpenoids backbone biosynthesis (20 unigenes involved), and fatty acid biosynthesis (26 unigenes involved).

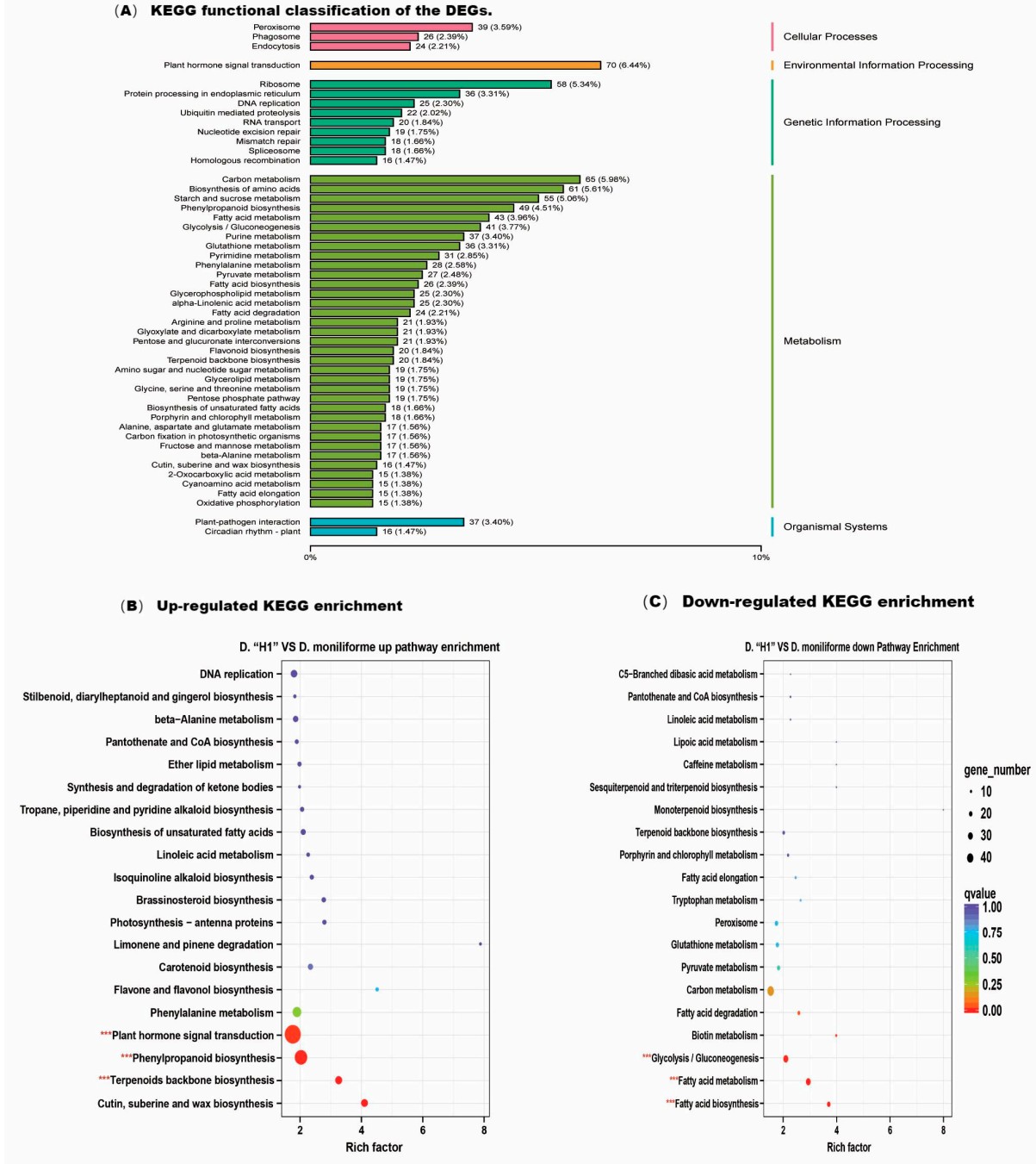

**Figure 4.** KEGG functional classifications and enrichment analysis of the DEGs. The *** in the figure indicate the statistical significance (*p* < 0.01) of the two *Dendoubium* species.

### 3.5. Protein-Protein Interaction (PPI) Network Analysis of the DEGs

PPI network analysis of a total of unigenes 95 DEGs co-regulated by phenylpropanoid biosynthesis, terpenoids backbone biosynthesis, and the fatty acid biosynthesis pathway was performed based on the STRING database. The results showed that proteins of 15 DEGs interacted with each other, as shown in Figure 5, and four of the up-regulated genes were LIS genes. Detailed information and nucleotide sequences for these 15 genes are listed in Supplementary Table S3. Combined with the results of physiological volatile compounds, the content of LIS might be critical for fragrance formation.

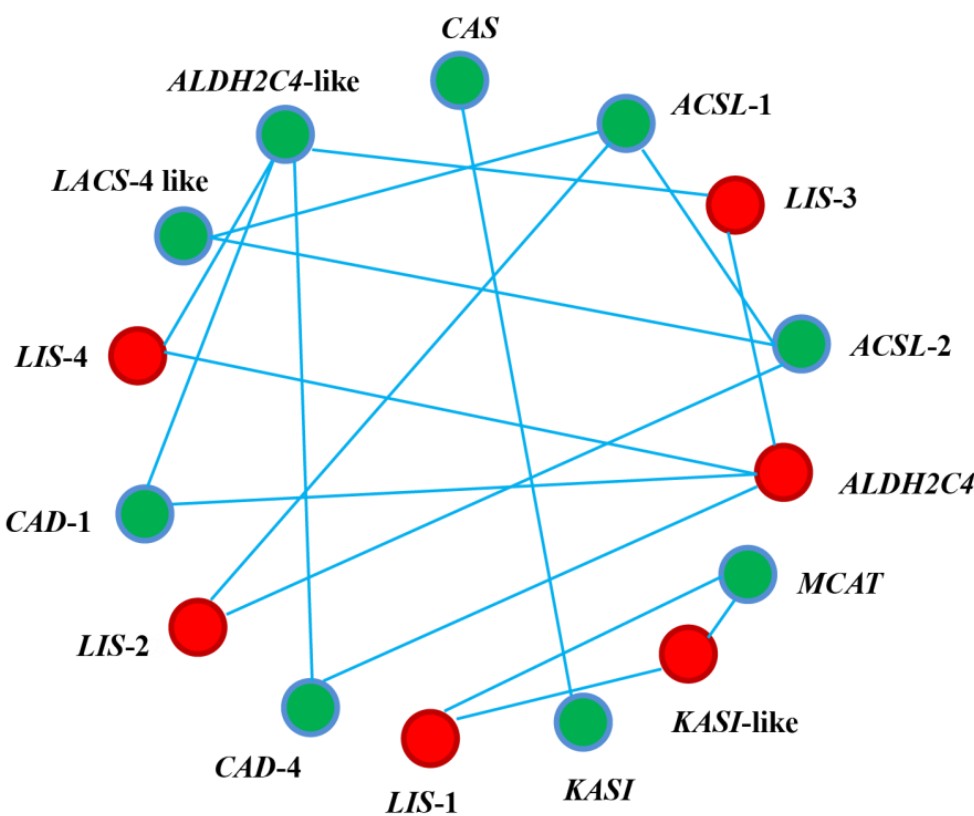

**Figure 5.** Protein-protein interaction network analysis of DEGs. Each node in the figure represents a protein of the gene, and the size of the nodes indicates the degree of correlation between them. Each line indicates the interaction between proteins. The green and red dots represent down-regulated and up-regulated, respectively.

### 3.6. Transcription Factor Prediction in Two Dendrobium Species

A total of 1563 transcription factors (TF) covering 66 TF families were identified from the transcriptome data of the two *Dendrobium* species. Family information on transcription factors is listed in Supplementary Table S4. C2H2, MYB, AP2/ERF-ERF, bHLH, and NAC families were significantly enriched, among which C2H2, MYB, and AP2/ERF-ERF families were dominant, with 117, 115, and 104 differentially expressed transcription factors, respectively (Figure 6). Our results suggested that these enriched TFs activated or inhibited downstream functional genes by regulating specific transcription factors during floral fragrance formation in *Dendrobium.*

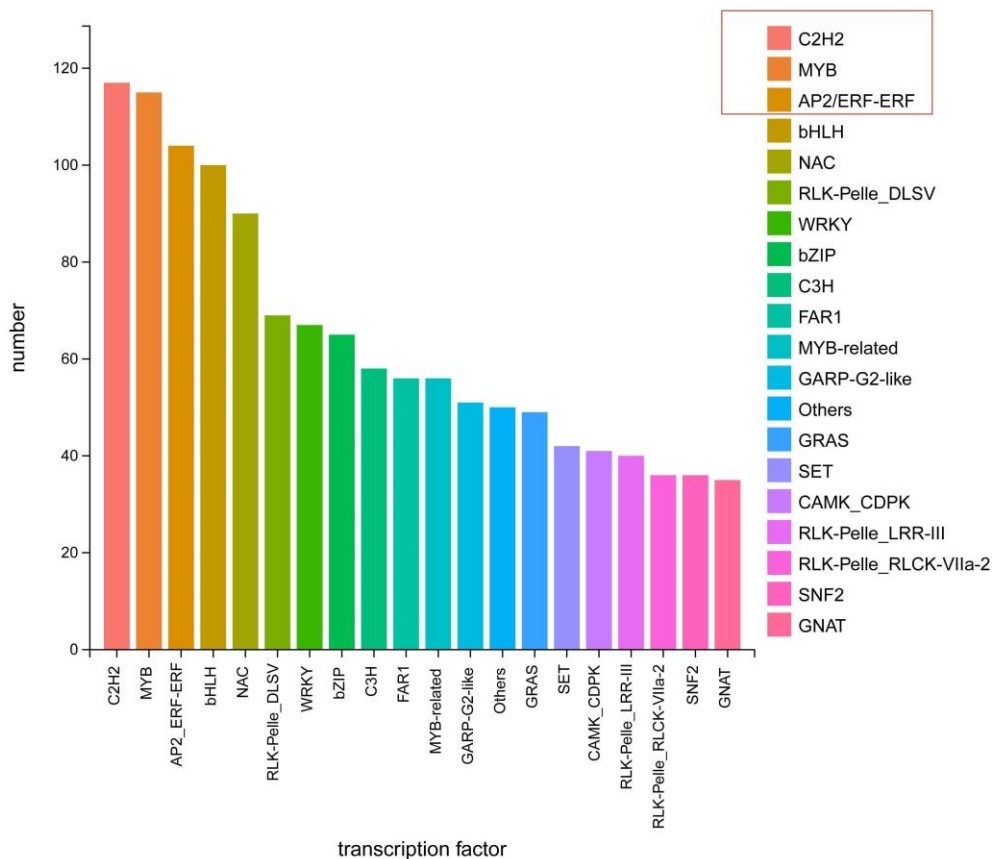

**Figure 6.** Transcription factor predictions.

*3.7. Validation of the DEGs Expression*

Fifteen DEGs associated with phenylpropanoid biosynthesis, terpenoids backbone biosynthesis, and fatty acid biosynthesis were selected for qRT-PCR analysis to verify the transcriptome data. These genes information as well as primers are shown in Supplementary Table S5. The qRT-PCR results showed the expression profiles of the selected genes were in agreement with the transcriptome data (Figure 7). The consistent expression profiles of qRT-PCR and transcriptome indicated the reliability of the transcriptome data. Interestingly, four of them were Linalool synthase (LIS) genes belonging to the terpene synthase family, which are associated with the monoterpene biosynthesis pathway and showed up-regulation in transcriptome and qRT-PCR expression. These results suggest that the expression level of LIS genes may play a crucial role in the genetic regulation mechanism of floral formation and synthesis.

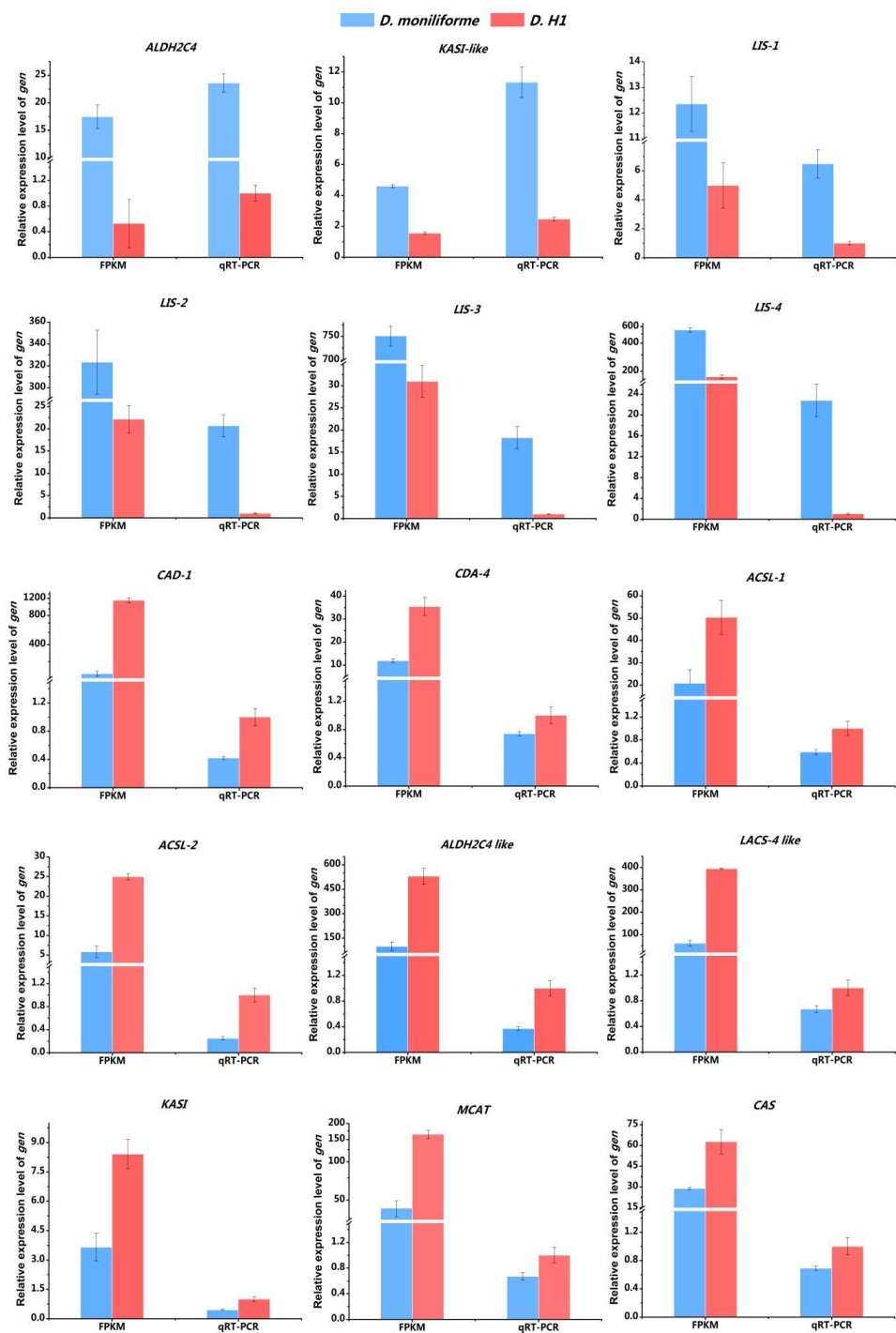

**Figure 7.** Relative expressions of 15 genes at the transcriptome and mRNA levels.

## 4. Discussion

There were significant differences in the volatile components and relative contents of different *Dendrobium* species, and the main volatile components played a decisive role in their fragrance. For instance, esters are the aroma volatile components of *D. lohohense*, while in *D. densiflorum* are mainly alkanes. The volatile content of linoleic acid and linolenic acid accounted for more than 60% of the total content in *D. thyrsiflorum* [48]. Geraniol and linalool are major floral scent compounds in *D. hancockii* [49]. The main volatile terpenoids in *D. chrysanthum* flowers were α-phellandrene, α-pinene and α-thujene [50]. In *D. officinale*, different cultivars with different smells had different contents of volatile compounds such as α-thujene, linalool, and α-terpineol [29,51]. In this experiment, SPME and GC-MS

were used to extract and detect the volatile components of flowers. Although there were certain components and content differences in different florescences and parts, 60 and 52 volatile compounds were collected from *D. moniliforme* and *D.* "H1", respectively. These compounds, according to their main chemical group, were classified as terpenes, aromatics, fatty acids, nitrogenous compounds, ketone, alcohols, and alkanes. The results showed that terpenoids were the main components of aromatic compounds in the two species, and their relative contents were 76.69% (*D. moniliforme*) and 79.09% (*D.* "H1") at full bloom stage, respectively. In petals, the contents of terpenoids in *D. moniliforme* and *D.* "H1" accounted for 77.25% and 50.38%, respectively. Among terpenoids, α-Pinene is the most abundant compound in the four florescences and different parts of the two species flowers. At the same time, we found that the contents of volatile terpenoids in two *Dendrobium* species were significantly different. For example, linalool was detected in four flowering stages in *D. moniliforme*, with the content ranging from 10.21% to 27.84% and reaching 30.86% in the petals. In contrast, *D.* "H1" can only be detected in three flowering stages, with the content ranging from 0.44% to 1.37% and only 2.73% in the petals. We hypothesized that LIS content is one of the reasons for the difference in scent between the two species.

Plant volatile terpenes play critical roles in the formation of orchid floral scents, which include monoterpenes, sesquiterpenes, and diterpenes [52]. In the current study, up-regulated KEGG enrichment pathway analysis showed that the terpenoids backbone biosynthesis pathway was highly enriched in *D. moniliforme*. The findings reinforce that terpenoids biosynthesis increased in high-scent *Dendrobium*. This study suggests a consistent relationship between the *Dendrobium* of floral scents and terpenoids biosynthetic pathways. It is urgent to study the differential gene functions of these two species.

In addition to terpenoids, the floral scents of *Dendrobium* contain a series of secondary metabolites such as benzenes/phenylpropanes, fatty acid derivatives, sulfur, or nitrogen compounds released by flowers [53]. Thus, a total of 95 genes related to these pathways in KEGG were selected for PPI network analysis to clarify the functional network that controls Aroma formation. The results showed that there were 15 closely connected genes, among which 4 were LIS genes, and their expressions were up-regulated. Real time PCR analysis of the 15 closely related genes showed that their expressions were completely consistent with the transcriptome data, indicating that the transcriptome data used in this study had strong credibility.

As an important member of the monoterpene family, linalool has a unique fragrance and various biological properties, which show important application value in attracting predators and repelling pests [54,55]. For example, Arabidopsis plants release linalool by transferring the TPS gene to repel aphids [56]. Additionally, spraying inducible (3S)-linalool on OsLIS-silenced rice plants could repel infestation by the rice brown planthopper Nilaparvata lugens [57]. TPS family genes that catalyze the production of linalool have also been found in *freesia hybrids* [58], *Malus domestica* [59], and *Vitis vinifera* [60].

TPS genes are mainly responsible for the production of volatile terpenes in plants [52]. In the present study, the LIS genes we found in *D. moniliforme* and *D.* "H1" that synthesize linalool belong to the TPS gene family. TPS is the primary enzyme that catalyzes the formation of linalool from the substrate GPP. In orchid flowers, the high volatility of terpenoids promotes floral fragrances. It was confirmed that linalool and geraniol were the main floral compounds in *Phalaenopsis bellina* [61]. In in vitro experiments, the TPS10 gene uniquely converted GPP to linalool in *D. officinale* [24]. In our present research, combined analysis of volatile compounds and transcriptomes in two *Dendrobium* species revealed that four LIS genes belonging to the TPS family, which were selected from the terpenoid biosynthesis, were involved in the positive regulation of floral formation in high-scent species. We speculate that inducing specific LIS gene expression can increase the production of linalool, which promotes the formation of flower fragrance. These results deepen our understanding of the molecular mechanism of *Dendrobium* flowers. It is valuable to further study the function of these genes, and these findings will provide a valuable reference about the terpene biosynthetic pathway in orchids.

## 5. Conclusions

There are many species of *Dendrobium*, but few have flowers that are both fragrant and beautiful. By combining volatile constituent determination and transcriptome sequencing analysis, we compared floral components and genes of the high-scent species *Dendrobium moniliforme* (*D. moniliforme*) and light-scent species *Dendrobium* "H1" (*D.* "H1"). Physiological results showed that the terpenoids content (especially linalool content) in different species is closely related to floral fragrance, and the petals may be the most important part affecting the release of volatiles. In the transcriptome analysis results, 5790 DEGs were identified in *D.* "H1" and *D. moniliforme*. These DEGs were highly enriched in phenylpropanoid biosynthesis (49 genes), fatty acid biosynthesis (26 genes), and terpenoid backbone biosynthesis (20 genes). Using them to construct the PPI network, four LIS (linalool) genes in the terpenoids biosynthesis pathway were closely connected, and all their expressions were shown to be up-regulated. The results indicated that the linalool biosynthesis in terpenoids was closely related to the formation of floral fragrances in *D. moniliforme* species. The key DEGs related to LIS metabolism screened in this study suggested that they played important regulatory roles in floral synthesis and release. The future work should focus on confirming the function of these genes and providing ideas for the cultivation of new aromatic species.

**Supplementary Materials:** The following supporting information can be downloaded at: https://www.mdpi.com/article/10.3390/horticulturae9070745/s1, Table S1: Changes of the main scent compositions and relative content in *Dendrobium moniliforme*; Table S2: Changes of the main scent compositions and relative content in *Dendrobium* "H1"; Table S3: Detailed information and nucleotide sequences of PPI network genes; Table S4: Family information of transcription factors; Table S5: Primers for qRT-PCR.

**Author Contributions:** S.Q. initiated, designed the experiment and revised the manuscript. Y.Y., K.X., Q.W., X.L., S.L. and Z.Z. performed the experiments and collected the data. Y.Y. analyzed the data and wrote the manuscript. All authors have read and agreed to the published version of the manuscript.

**Funding:** This work was funded by the National Natural Science Foundation of China (31560567), the Hainan Natural Science Fund Project (320RC722), the National Natural Science Foundation of Guangxi (2020GXNSFAA297260), the Start-up Fund of Innovation Team of Guangxi Academy of Sciences for Innovation and Utilization of Germplasm in Horticultural Crops (CQZ-E-1919), the Guilin Innovation Platform and Talent Plan (20210102-3), the Fundamental Research Fund of Guangxi Institute of Botany (23011), and the fund of the Guangxi Key Laboratory of Plant Functional Phytochemicals and Sustainable Utilization (ZRJJ2023-1).

**Data Availability Statement:** All sequencing data are available through the NCBI Sequence Read Archive under the accession number PRJNA976822.

**Conflicts of Interest:** The authors declare no conflict of interest.

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
