# Peer review of "Combined Analysis of Volatile Compounds and Extraction of Floral Fragrance Genes in Two Dendrobium Species"

_horticulturae, doi:10.3390/horticulturae9070745_

Round 1

Reviewer 1 Report

Horticulture

Article: “Combined analysis of volatile compounds and extraction of floral fragrance genes in two Dendrobium species”

-        - “Dendrobium belongs to orchidaceae”;I suggest that is better to use this term (Dendrobium genus) or (species of Dendrobium genus).

-         -  “Among them, terpenoids are one of the most abundant and diverse aromatic 43compounds in flowers”. add references.

-          - “All terpenes are synthesized by terpene synthases (TPSs), which is the last link in the process of terpenoids synthesis”. This is not clear.

-          - “TPSs have conserved regional structure DDXXD in plants”. What is the meaning of DDXXD.

-          - “Structural differences may lead to new catalytic activities, resulting in the formation of many different species of volatile terpenoids from the same substrate”. Which classes of terpenoids contain  volatile compounds?

-         -  “In this experiment, the flowers of ligh-scent D. ‘H1’ and high-scent D. moniliforme were used as materials”. Add the full scientific name, and add the author name of the taxonomy of D. moniliforme. Explain what is this name, or why is this specie unidentified: D. ‘H1’.

-          - “According to the SPME and GC-MS analysis of volatile compounds from D. moniliforme and D. ‘H1’, a total of seven compounds including terpene, aromatic, fatty acids, nitrogenous compound, ketone, alcohols and alkanes were collected”. This is not clear or not logic. These identified natural products are not 7 compounds, these are main chemical groups. Example terpenes are main chemical group, while contains subclasses, some of them contain volatile compounds. Correct such phrase across the manuscript.

-        -   Table 1. This table contains general chemical groups but no specific volatile compounds. Such general groups can not give any specific data about the volatile properties of the identified flowers.

Moderate editing of English language required

Author Response

Thank you very much for your kindly review and these helpful comments. 

The main corrections in the paper and the responds to the comments are in the appendix file, we hope the modifications could meet with your approval.

Reviewer 2 Report

Line 11; please write the full scientific name of the plant.

line 90; It is not known whether you have used fresh materials or dried ones since you have shown in figure 1 as cultivated.

line 108; why you did not use steam distillation for the extraction of essential oils?

line 179-180; does not make sense please rewrite these sentences. 

The identification of compounds is not clear in the methodology so please describe it.  

You have used too many figures in the manuscript please try to keep balance and you can bring tables from supplementary and replace them with figures.  

What is the difference between Table 1 and Table S1, S2? I think tables S1 and S2 can be merged and replaced to Figure 1 and Table 1. For figures 3-7 also you can improve them by deleting unnecessary figures.

Author Response

(The authors gave the same response as above.)

Reviewer 3 Report

See the report of comments

See the report

Author Response

(The authors gave the same response as above.)

Round 2

Reviewer 1 Report

Article: “Combined analysis of volatile compounds and extraction of floral fragrance genes in two Dendrobium species”

1.       “The main volatile compounds of Dendrobium included terpenoids (also known as isoprene)”. Isoprene is the building block (unit) of the various types of terpenes or terpenoids.

2.       “TPS is positioned at the branch point of the isoprenoid pathway, which is the last link in the synthesis process of terpenoids. In addition, TPS harbors conserved regional structure such as DDxxD (an aspartate-rich motif that interacts with divalent metal ions involved in positioning the substrate for catalysis ) in plants. This phrase must be improved. Isoprene is intermediate biosynthetic precursor for all teprenes and terpenoids.

Minor editing of English language required

Author Response

Dear Reviewer

  Thank you very much for reviewing and revising the details again. The main corrections in the paper and the responds to the comments are revised. Please see the attachment. 
